# Ethyl pyruvate protects SHSY5Y cells against 6-hydroxydopamine-induced neurotoxicity by upregulating autophagy

**Yuening Luo, Kazuichi Sakamoto**⊙*

Faculty of Life and Environmental Sciences, University of Tsukuba, Tsukuba, Ibaraki, Japan

* sakamoto@biol.tsukuba.ac.jp

**Data Availability Statement:** All relevant data are within the paper and its Supporting Information files.

## Abstract

Parkinson disease is a chronic progressive neurodegenerative disorder with a prevalence that increases with age. The glycolytic end-product pyruvate, has antioxidant and neuroprotective feature. Here, we investigated the effects of ethyl pyruvate (EP), a pyruvic acid derivative, on 6-hydroxydopamine-induced SH-SY5Y cell apoptosis. Ethyl pyruvate decreased protein levels of cleaved caspase-3, phosphorylated endoplasmic reticulum kinase (pERK), and extracellular signal-regulated kinase (ERK), suggesting that EP reduces apoptosis *via* the ERK signaling pathway. Ethyl pyruvate also decreased oxygen species (ROS) and neuromelanin contents, suggesting that it suppresses ROS-mediated neuromelanin synthesis. Furthermore, increased protein levels of Beclin-1 and LC-II, and LC-II:LC-I ratios indicated that EP upregulates autophagy.

## Introduction

Parkinson disease (PD) is the second most common progressive neurodegenerative disease after Alzheimer disease [1]. Parkinson disease is closely associated with age; it affects ~ 0.1% of the global population and increases to 1% and 2%–4% of the those aged over 60 and 80 years, respectively [2, 3]. Parkinson disease is characterized by a loss of dopaminergic neurons in the substantia nigra pars compacta (SNpc), striatal dopamine depletion, and abnormal α-synuclein aggregation [4, 5]. Other manifestations such as mitochondrial dysfunction, neuroinflammation, and oxidative stress have been identified in models of PD [6–8].

The role of α-synuclein aggregation and oxidative stress in PD pathogenesis has been established [5, 9]. Alpha synuclein is easily misfolded and polymerized, and unlike unfolded proteins, the primary degradation of aggregated α-synuclein is autophagy [10–12]. Rapamycin increases autophagy and protects against PD in models [11]. These findings indicate that autophagy might function as a neuroprotective strategy for PD by eliminating aggregated α-synuclein.

Oxidative stress is a risk factor for PD. Oxidative stress damages proteins, nucleic acids, and lipids that leads to cell dysfunction and death [7]. Numerous compounds with antioxidant activities have potential for treating PD according to findings *in vitro* and *in vivo* [13, 14].

The dopaminergic neurotoxin 6-hydroxydopamine (6-OHDA) is selectively uptaken by dopaminergic neurons *via* the dopamine transporter (DAT) and stored in mitochondria,

**Funding:** The author(s) received no specific funding for this work.

**Competing interests:** The authors have declared that no competing interests exist.

where it promotes the generation of free radicals and decreases ATP synthesis [15]. It is widely used to develop PD models *in vivo* and *in vitro* [16, 17]. It is easily oxidized to generate the superoxide anion, para-quinone, as well as hydrogen peroxide [18]. This is followed by intracellular reactive oxygen species (ROS) that eventually cause neuronal cell death [19]. The major advantage of the PD model induced by 6-OHDA is that the range of dopaminergic lesions can be controled and motor deficits can be quantified [20].

The final product of glycolysis is pyruvate, an anti-oxidant and ROS scavenger *in vitro* and *in vivo* [21, 22], but it is unstable in aqueous solution, which limits its application as a therapeutic agent. In contrast, the ethyl ester form of pyruvate, ethyl pyruvate (EP), is relatively stable and also has anti-oxidative, anti-inflammatory, and anti-apoptotic properties [22–24]. Ethyl pyruvate prevents the degeneration of nigrostriatal dopamine (DA) neurons, increases striatal dopamine levels, and improves motor function in PD models *in vivo* [25]. It also protects rat pheochromocytoma PC12 cells from dopamine-induced cytotoxicity *in vitro* [24].

However, the protective effect of EP on 6-OHDA induced SH-SY5Y cell death remains unclear. The present study aimed to determine whether EP is neuroprotective against 6OHDA-induced cell death i*n vitro* and evaluated its potential for treating PD.

## Materials and methods

### Cell culture

The human neuroblastoma cell line SH-SY5Y (Riken Cell Bank, Tsukuba, Japan) was cultured and maintained in 5% $CO_2$ at 37˚C in DMEM/F12 medium (Sigma-Aldrich Corp., St. Louis, MO, USA) supplemented with 10% FBS (Hyclone Laboratories, Logan, UT, USA), penicillin (Wako, Tokyo, Japan), and streptomycin (Wako, Tokyo, Japan). Cells ($6.25 \times 10^4$ cells/cm$^2$) were seeded in 96-well plates and 6-cm dishes for 24 h, then incubated with 1, 2.5, and 5 mM EP (Wako, Tokyo, Japan) for 24 h followed by 75 μM 6-OHDA (50 mM; Sigma-Aldrich Corp.) dissolved in (dDW) for 6 h to assay apoptosis or 24 h for MTT assays, western blotting and melanin quantitation.

### MTT assays

Cytotoxicity was determined using 3-(4,5-dimethylthiazol-2-yl)-2,5-diphenylterazolium bromide (MTT) assays, (Sigma-Aldrich Corp.) as described [26]. The medium was replaced with 90% DMEM/F12 containing 10% MTT and incubated at 37˚C for 4 h. Thereafter, 10% SDS was added and the cells were incubated overnight at room temperature (RT). Absorbance at 570 nm was determined using a microplate reader (BioTek, Tokyo, Japan).

### Apoptosis assays

SH-SY5Y cells cultured in DMEM/F12 medium were seeded for 24 h, then incubated with various concentrations of EP for 24 h followed by 6-OHDA for 6 h. Apoptotic, necrotic, and live cells were quantified using apoptosis/necrosis detection kits (blue, green, red, respectively; Abcam, Cambridge, UK) as described by the manufacturer. Fluorescence microscopy (Keyence, Tokyo, Japan) images were acquired and fluorescence intensity was quantified using ImageJ to determine apoptosis.

### Assays of ROS

SH-SY5Y cells were cultured in DMEM/F12 medium, then incubated with 1, 2.5, or 5 mM EP for 24 h followed by 75 μM 6-OHDA for 6 h. Intracellular ROS were determined using DCFDA/H2DCFDA cellular ROS assay kits (Abcam, Cambridge, UK) as described by the

manufacturer. Fluorescence microscopy (Keyence) images were acquired and fluorescence intensity was quantified using ImageJ to determine ROS.

## Melanin quantitition

The cells were sedimented by centrifugation, trypsinized, suspended in 1 N NaOH, then heated at 80°C for 1 h. Absorbance at 405 nm was detemined using a microplate reader (Bio-Tek). The final relative melanin content was normalized to the total protein content measured using BCA assay kits (Thermo Fisher Scientific Inc., Waltham, MA, USA).

## Western blotting

Cells sedimented by centrifugation were washed twice with cold PBS, then lysed using RIPA buffer (150 mM NaCl, 1 mM EDTA, 50 mM Tris-HCl, 10 mM NaF, 1 mM $Na_3VO_4$, 1% Triton X-100, 0.1% sodium dodecyl sulfate (SDS), 0.5% Na-deoxycholate, and protein inhibitor). Proteins were then resolved by SDS-polyacrylamide gel electrophoresis (SDS-PAGE) using 4%–20% polyacrylamide gels. The resolved proteins were blotted onto PVDF membranes and incubated with the following primary antibodies diluted 1:1,000 (Beclin (#3495), caspase-3 (#9662), Phospho-p44/42 MAPK (Erk1/2) (Thr202/Tyr204), (pERK) (#9101), p44/42 MAPK (Erk1/2; ERK (#9102), and Microtubule-associated proteins 1A/1B light chain 3B (LC3 A/B) XP (#12741) (all from Cell Signaling Technology, Danvers, MA, USA). The membranes were incubated with horseradish peroxidase (HRP)-conjugated secondary antibody diluted 1:1,000 (#7074) (Cell Signaling Technology) at RT for another 1 h. The secondary antibodies were labeled using the chemiluminscent substrate LumiGLO® (Cell Signaling Technology) and proteins were detected using the AE-9300 Ez-capture MG system (Atto Corporation, Tokyo, Japan) and quantified using ImageJ software (National Insititues of Health [NIH], Bethesda, MD, USA).

## Statistical analysis

The results are expressed as the means ± SD of at least three experiments. Data was compared among groups by ANOVA post-hoc tests using SPSS Statistics (SPSS Inc., Chicago, IL, USA). Values with $p < 0.05$ were considered statistically significant.

# Results and discussion

## Ethyl pyruvate protected SH-SY5Y cells against 6-OHDA-induced cytotoxicity

MTT assay was using to measure the cell viability. Our results showed that 5mM EP have no cytotoxic; whereas 10 mM EP was clearly cytotoxic compared with control cells (Fig 1A). Therefore, we applied < 5 mM EP in further experiments. 75 μM 6-OHDA showed the strong cytotoxicity compared to that of the control group, However, Ethyl pyruvate pretreat group showed a protective effect against 6-OHDA-induced cytotoxicity (Fig 1B).

## Ethyl pyruvate decreased SH-SY5Y cell apoptosis induced by 6-OHDA

Apoptosis assay was used to measure the effect of Ethyl pyruvate on apoptosis. 6-OHDA treat group showed a significant increase in cell death compared to that in the control group (Fig 2A). Then, the protein expression levels of caspase-3 were also measured. Similar results as apoptosis, cleaved caspase-3 protein levels were increased in 6-OHDA group and EP-treated group shown a decreased compared to the 6-OHDA group (Fig 2B and 2C).

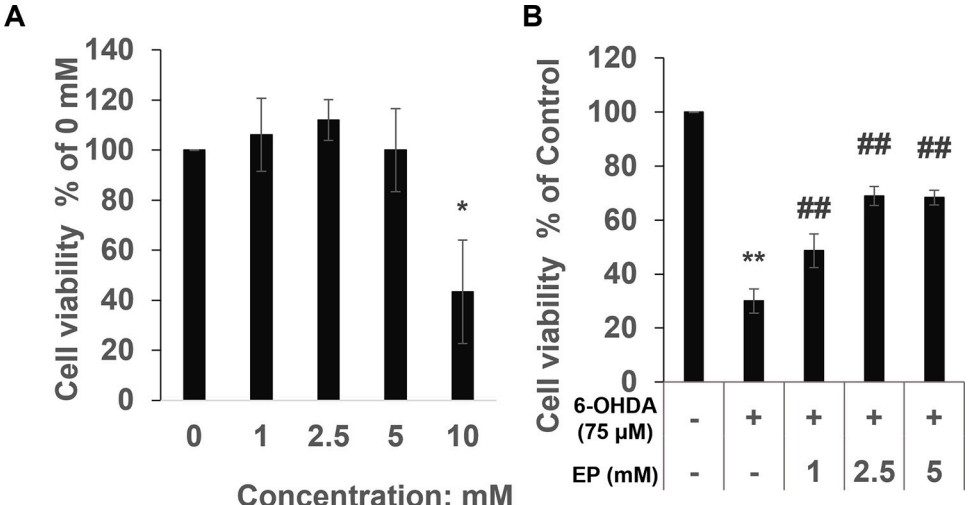

**Fig 1. Effects of EP on cytotoxicity induced by 6-OHDA.** (a) Cytotoxicity of EP. (b) Effects of EP on 6-OHDA-induced cell death determined as cell viability using MTT assays. Results are expressed as means ± standard deviation (N ≥ 3), $^*$P < 0.05, $^†$P < 0.01 $vs.$ controls; $^‡$P < 0.01 $vs.$ 6-OHDA.

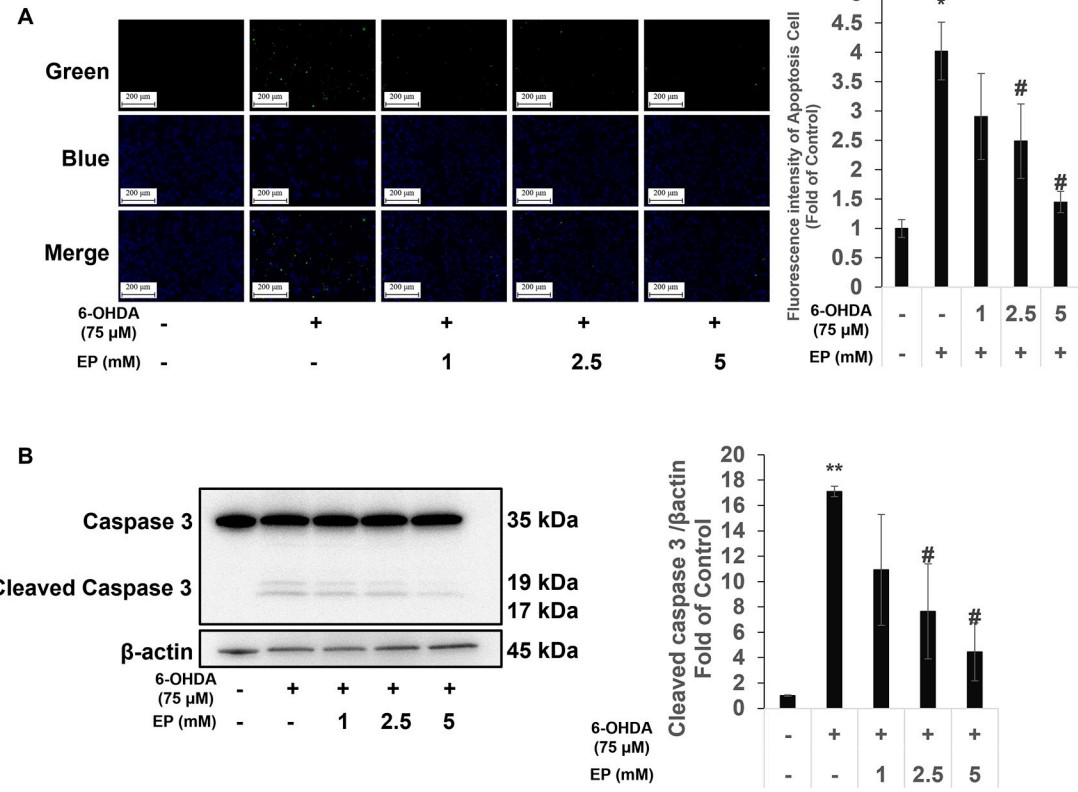

**Fig 2. Effects of EP on 6-OHDA induced cell death.** (a) Apoptotic or necrotic (green), and live (blue) cells visualized using fluorescence microscopy. (b) Representative western blots of caspase-3 and cleaved caspase-3 (c) quantified using ImageJ. Results are expressed as means ± standard deviation (N ≥ 3. $^*$P < 0.01 $vs.$ control; $^†$P < 0.05 $vs.$ 6-OHDA. Scale bars, 200 μm.

## Ethyl pyruvate reduced 6-OHDA-induced ROS levels in SH-SY5Y cells

ROS assay was performed to evaluate the effect of EP on ROS levels. 6-OHDA treat group showed a significant increase in ROS levels compared to the control group; EP treat group showed a decreased in ROS level compared to 6-OHDA group (Fig 3A).

Then, protein levels of pERK and ERK, were also measured. The pERK/ERK ratio increased in the 6-OHDA treat group compared to the control group and EP treat group showed a decreased in pERK/ERK ratio (Fig 3B and 3C).

## Ethyl pyruvate decreased neuromelanin in SH-SY5Y cells induced by 6-OHDA

To evaluate the effect of EP on neuromelanin production, melanin content was measured. 6-OHDA treat group showed a significant increase in neuromelanin compared to the control group. EP treat group showed a decreased in neuromelanin content (Fig 3D and 3E).

## Effects of EP on autophagy-related gene expression in SH-SY5Y cells

We measured protein levels of the autophagy-related genes *Beclin-1* and *LC3*. Protein levels of Beclin-1 decreased, whereas LC3-II did not significantly change in the 6-OHDA, compared with controls. Ethyl pyruvate increased protein levels of Beclin-1 and LC3-II compared with

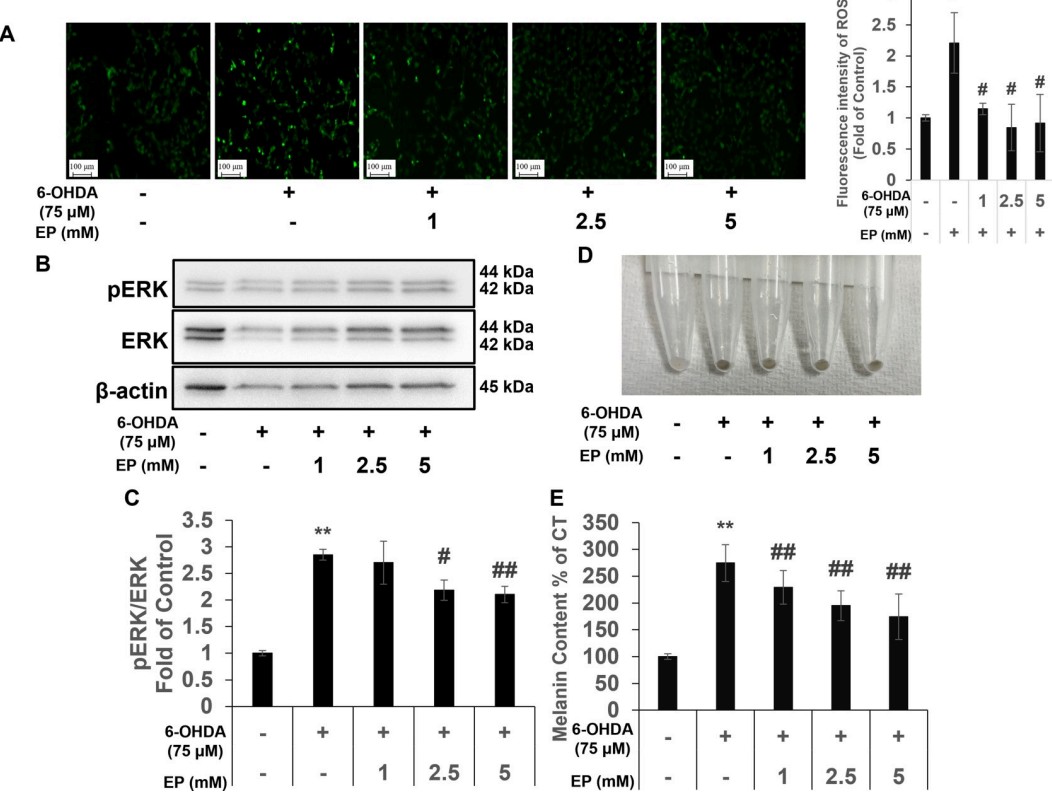

**Fig 3. Effects of ethyl pyruvate (EP) on 6-hydroxydopamine (6-OHDA)-induced intracellular reactive oxygen species (ROS) and neuromelanin contents.** (a) Reactive oxygen species (ROS) determined from fluorescence microscopy images. Scale bars: 100 μm. Representative western blots (b) of pERK and extracellular signal-regulated kinase (ERK) quantified (c) using ImageJ. (d) Intracellular neuromelanin deposition. (e) Neuromelanin content and normalized the controls. Results are expressed as means ± standard deviation (N ≥ 3) *P < 0.01 *vs.* control; †P < 0.05 and ‡P < 0.01 *vs.* 6-OHDA.

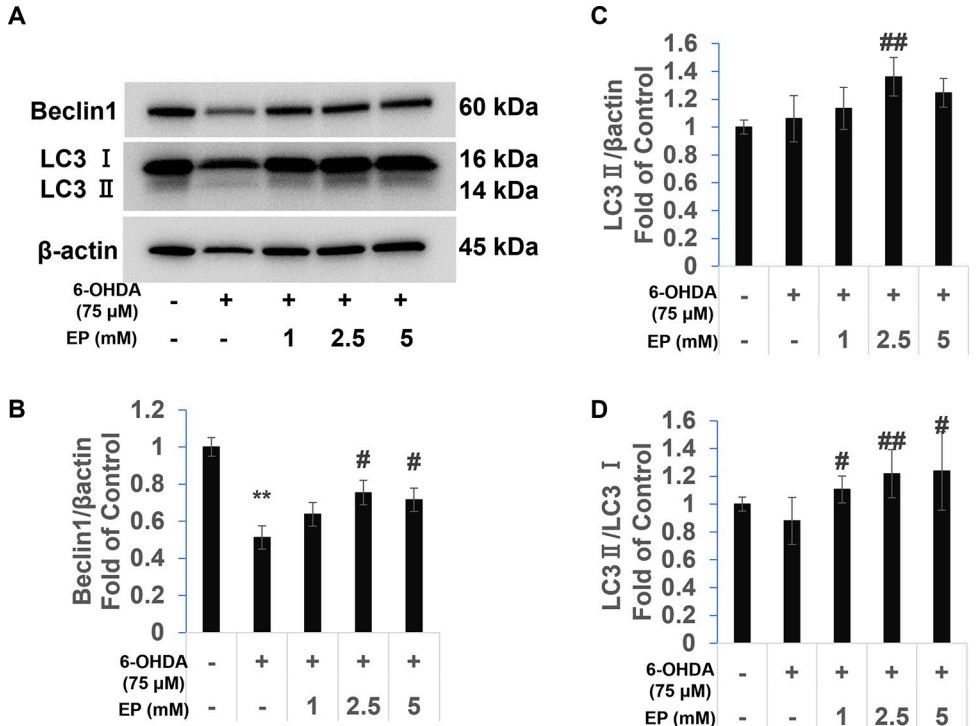

**Fig 4. Effects of EP on autophagy-associated Beclin-1 and LC3 gene expression.** (a) Representative western blots of LC3 and Beclin-1. Protein bands of (b) Beclin-1 and (c) LC3-II quantified by ImageJ. (d) Relative ratios of LC3-II/ LC3-I band density. Results are expressed as means ± standard deviation (N ≥ 3) *P < 0.01 *vs.* control; †P < 0.05 and ‡P < 0.01 *vs.* 6-OHDA).

the 6-OHDA group (Fig 4A–4C). The LC3-II/LC3-I ratio (marker of autophagy) did not significantly differ between the 6-OHDA and control groups. However, EP increased the LC3-II/ LC3-I ratio compared with the 6-OHDA group (Fig 4D). These results suggested that EP upregulates autophagy.

Neuronal cells are susceptible to oxidative stress-induced cell damage that leads to neuronal cell death and is a risk factor for age-related neurodegenerative diseases [7]. Neurotoxic 6-OHDA can be easily oxidized to produce intracellular ROS and is used to destroy dopaminergic neurons in the brain [19]. The pyruvic acid derivative EP is an endogenous antioxidant metabolite [22]. The present study aimed to determine the effects of EP on 6-OHDA-induced neuronal death.

Differentiated SH-SY5Y cells might not be suitable for 6-OHDA-induced PD models due to their high tolerance of 6-OHDA toxicity [27]. Therefore, we investigated the effects of EP in undifferentiated SH-SY5Y cells.

We found that EP (< 5 mM) significantly increased the viability of cells incubated with 6-OHDA but did not affect that of control cells, suggesting a protective effect against 6-OHDA-induced cytotoxicity.

The results of apoptosis assays confirmed the effect of EP on 6-OHDA-induced cell death. Apoptosis rates increased in cells incubated with 6-OHDA, which was consistent with previous findings [28, 29], and EP decreased them. We then measured levels of caspase 3 and cleaved caspase 3 proteins. The increased abundance of cleaved caspase-3 in the 6-OHDA group, which was consistent with our previous findings [29], was reduced by EP, confirming the protective effect of EP against 6-OHDA-induced cell death.

A significant role of oxidative stress in PD has been identified [7, 30]. Antioxidant compounds such as asiaticoside and Ginsenoside-Rg1 are promising candidates for treating PD [13, 14]. The present and our previous [29] findings revealed EP decreased the elevated ROS levels in cells incubated with 6-OHDA.

Extracellular signal-regulated kinases (ERK1/2) are crucial regulators of neuronal responses associated with cell death [31, 32], and their activation also plays important roles in several models of 6-OHDA-induced cell death [33, 34]. Cell death induced by 6-OHDA involves ERK activation [33], but not SAPK/JNK or p38 kinase [35]. Ethyl pyruvate attenuates p-ERK expression in formalin-induced neuronal models [36] and inhibits ERK phosphorylation in those of LPS-induced inflammation [37].

Therefore, we measured levels of pERK and ERK proteins and found significantly increased ERK phosphorylation in cells incubated with 6-OHDA, which was consistent with previous findings [29, 31]. Furthermore, EP decreased ERK phosphorylation, indicating that it reduces apoptosis *via* the ERK signaling pathway.

Neuromelanin is produced in human SNpc dopaminergic neurons over a lifetime and accumulates with age until it occupies most of the neuronal cytoplasm [38]. Exceeding the threshold amount of accumulated NM is associated with an age-dependent PD phenotype, and enhanced lysosomal proteostasis can reduce intracellular neuromelanin and prevent neurodegeneration [39]. Oxidative stress is also associated with neuromelanin synthesis [32]. We previously showed that EP decreased intracellular neuromelanin levels increased by 6-OHDA [40]. This suggested that EP suppresses ROS-mediated neuromelanin synthesis.

Abnormal α-synuclein aggregation is also a feature of PD [5]. Autophagic proteolysis is an important degradation pathway for α-synuclein, and that increased autophagy is also relative to cell survival [10, 11]. Autophagy blocked with chloroquine induces increased α-synuclein accumulation, whereas autophagy activation by rapamycin results in α-synuclein clearance [41]. These findings showed that autophagy might play an important role in PD therapy. One study found that 6-OHDA can cause cell apoptosis, decrease autophagy markers (LC3-II/LC3-I, Beclin-1) and increase phosphate mTOR/mTOR [42]. The mTOR inhibitor rapamycin can restore increased mTOR activity caused by overexpressed α-synuclein [43], and the autophagy inhibitor chloroquine can block this protect effect [42]. Both MPTP and 6-OHDA increase α-synuclein [35, 44], and consequently inhibit autophagy [45], whereas Ethyl pyruvate decreases α-synuclein abundance [46].

We measured levels of the autophagy-related proteins, Beclin-1, and LC3. We found that 6-OHDA significantly decreased Beclin-1 expression but did not significantly alter LC3-II levels, which was consistent with our previous findings [29]. Ethyl pyruvate significantly increased Beclin-1 and LC3-II protein levels. These results suggested that EP protects SH-SY5Y cells against 6-OHDA-induced cell apoptosis by upregulating autophagy. However, 5 mM EP did not induce any significant differences in levels of Beclin-1 and LC3-II proteins compared with 2.5 mM EP, indicating that the antioxidant capacity of EP in cells is limited.

## Conclusions

We showed that EP reduced apoptosis, ROS, and neuromelanin levels, and improved autophagy. We believe that further therapeutic interventions targeting EP might prove beneficial and improve the etiology of neurodegeneration.

## Supporting information

**S1 Raw data.**
(ZIP)

## Acknowledgments

We afre gratefull to the partical support from the University of Tsukuba, Japan.

## Author Contributions

**Conceptualization:** Kazuichi Sakamoto.

**Data curation:** Yuening Luo, Kazuichi Sakamoto.

**Formal analysis:** Yuening Luo, Kazuichi Sakamoto.

**Funding acquisition:** Kazuichi Sakamoto.

**Investigation:** Yuening Luo, Kazuichi Sakamoto.

**Methodology:** Yuening Luo.

**Project administration:** Kazuichi Sakamoto.

**Resources:** Kazuichi Sakamoto.

**Supervision:** Kazuichi Sakamoto.

**Validation:** Yuening Luo, Kazuichi Sakamoto.

**Visualization:** Yuening Luo.

**Writing – original draft:** Yuening Luo.

**Writing – review & editing:** Yuening Luo, Kazuichi Sakamoto.

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
