## [Decision Letter · Decision Letter 0]

28 Sep 2022

PONE-D-22-21944Ethyl pyruvate protects SHSY5Y cell line against 6-hydroxydopamine-induced neurotoxicity by upregulation Autophagy.PLOS ONE

Dear Dr. Sakamoto,

Thank you for submitting your manuscript to PLOS ONE. After careful consideration, we feel that it has merit but does not fully meet PLOS ONE’s publication criteria as it currently stands. Therefore, we invite you to submit a revised version of the manuscript that addresses the points raised during the review process.

We look forward to receiving your revised manuscript.

Kind regards,

Khuen Yen Ng, PhD

Academic Editor

PLOS ONE

Journal Requirements:

 "NO." 

"This work was supported in part by the Grants-in-Aid for Scientific Research and Education from the University of Tsukuba, Japan."

"NO"

Reviewers' comments:

Reviewer's Responses to Questions

**Comments to the Author**

1. Is the manuscript technically sound, and do the data support the conclusions?

Reviewer #1: Partly

Reviewer #2: Yes

2. Has the statistical analysis been performed appropriately and rigorously? 

Reviewer #1: Yes

Reviewer #2: N/A

3. Have the authors made all data underlying the findings in their manuscript fully available?

Reviewer #1: Yes

Reviewer #2: No

4. Is the manuscript presented in an intelligible fashion and written in standard English?

Reviewer #1: Yes

Reviewer #2: No

5. Review Comments to the Author

Reviewer #1: The present manuscript by Luo and Sakamoto describes the neuroprotective effects of ethyl pyruvate (EP) against the 6-OHDA induced toxicity in SH-SY5Y cells, potentially mediated by the upregulation of autophagy. The current study appears similar to a recently published work by the same authors to some extent, including the in vitro model used, biochemical assays, presentation, etc. However, there are concerns in the manuscript that needs to be addressed and clarified.

1. Materials and Methods – Perhaps provide a detail description on how 6-OHDA and EP is dissolved/prepared prior to the treatment. Was there any solvent/vehicle control included in the cell viability assay?

2. Method and Materials – For the cell apoptosis (Section 2.3) and ROS (Section 2.4) assays, the authors mentioned a post-treatment of 6-OHDA for only 6 hrs, which is different from the MTT assay and melanin content measurement. Why is there a difference in the 6-OHDA treatment among the different assays? Is this a typo or did the authors perform prior screening assays (different concentration and time-point) to justify the extent of toxicity induced by 6-OHDA in SH-SY5Y cells?

3. Results – Section 3.2 – The authors mentioned a significant increase in cell death induced by 6-OHDA treatment, which is only presented in the photomicrographs. Perhaps the authors should present the quantification data, since this is also mentioned in the Methods and Materials (Pg. 4) to quantify the apoptotic, necrotic and live cells. Furthermore, the fluorescence images presented in Figure 2a appears to be of poor quality and dim throughout, which should not be the case.

4. Results – Section 3.3 – Similarly, measurement of the intracellular ROS levels should be presented instead of showing only the photomicrographs in the Results, as mentioned in the Methods section (Pg. 5).

5. There have been several prior studies on neuroprotective effects and mechanisms of EP in both in vitro and in vivo PD models. Perhaps the authors could improve the Introduction and/or Discussion of this manuscript to also discuss how this study fit into the current literature and strengthen the Discussion section.

6. Minor grammatical errors present throughout the manuscript and please check thoroughly (E.g., Pg.2, L. 1.; Pg.6, L.18; etc).

Reviewer #2: This is an interesting manuscript examining the effect of ethyl pyruvate (EP) on 6-hydroxydopamine-induced SH-SY5Y cell apoptosis. Authors stated that EP could decrease cleaved caspase-3, pERK, and ERK protein levels, suggesting that EP reduces apoptosis via the ERK signaling pathway. Moreover, this study demonstrated that EP increased the protein levels of Beclin-1, LC-II and the ratio of LC-II/LC-I, suggesting that EP can upregulating autophagy. This is totlly a "descriptive" observation. However, still I have some concerns about the methods and data.The authors would have to provide additional experimental evidences about MAPK signaling pathway and autophagy as well as have multiple rising questions. Authors should provide the answers of those questions.

Comments:

1. Details method should need to be clarified by the authors. Example in Fig 1B, authors used 6-OHDA and EP but in method section autors did not explain anything about their drugs and also concentrations. The drug and concentrations should also be stated in the method part.Moreover, in MTT method authors added 10 % SDS solution, author needs to explain the reason. Author should provide the ref. of this method.

2. In Fig. 3B, it seemed that pERK and ERK did not match with the quantitative values. There were some details that need to be clarified. The authors should provide the all raw protein bands of three independent experiments.

3. mTOR promotes anabolic metabolism and inhibits autophagy induction. Therefore, the regulation of autophagy with mTOR inhibitors provides a new therapeutic strategy for a variety of diseases. Thus, in this study author should observe the expression of mTOR complaxes such as mTORC1 and mTORC2 and also evaluate the effect of EP on mTORC1 and mTORC2.

4. Because authors focused on autophagy, thus to make a more influential study, authors should use Chloroquine and rapamycin in this study. Because Rapamycin promoted autophagy by blocking the mTOR pathway, and chloroquine enhanced apoptosis by blocking autophagy. Authors need to compare the effect of EP and Chloroquine and rapamycin on autophagy.

5. Authors should provide more information about 6-hydroxydopamine (6-OHDA) and Ethyl pyruvate (EP) in the introduction part. Example how 6-OHDA developes PD model? Are there any mechanism link with this disease?

6. Authors need to explain why they selected only ERK among the three proteins of MAPK pathway. How ERK plays an important role in this study? For more clarification authors should check p38 and JNK.

7. Authors stated that α-synuclein aggregation and oxidative stress in PD pathogenesis. Authors also stated the importance of α-synuclein in autophagy. Are there any reports about the α-synuclein and EP? If not, how authors explain this link between EP and α-synuclein?

10. The authors should indicate the specific cat. number and dilution factor of the antibodies used in the Methods section to improve the reproducibility of their findings.

11. Most of the images quality are very poor such as Fig 2A, 3A etc. Authors should provide high-quality images and blots.

12. Please check English grammar and sentence structure.

13. Authors need to use molecular weight in the western blot images.

14. In Fig 2 B cleaved caspase-3 showed 2 bands, Are there any reason, authors need to explain or provide another image?

15. In western blot images Beta actin should be equal but in Fig 2B,3B and 4A why in all groups beta-actins are not equal?

16. In western blot method, authors needs to explain about the dilution of antibodies. And also provide the how many percentage of gel authors used during WB?

17. In discussion part authors demonstrated that Extracellular signal-regulated kinases (ERK1/2) are crucial regulators of neuronal responses associated with cell death [26] [27]. In several different 6-OHDA-induced cell death model, ERK1/2 activation also plays an important role [28][29]. Therefore, we measured the protein levels of pERK and ERK. Cells exposed to 6-OHDA showed a significant increase in ERK phosphorylation consistent with previous studies[24][26], and pretreatment with EP decreased the ERK phosphorylation. This suggests that EP reduces apoptosis via the ERK signaling pathway. But authors did not explain how 6-OHDA -induced cell death model ? and how to make link with ERK. Additionally, authors should provide some ref. about the EP and ERK in other diseases. In overall author should re-arrage the some paragraphs of discussion part and make it more clear.

18. Apoptosis assay, ROS assay, Melanin Content first two lines are very similar, why? Authors needs to write in different ways. And in method section authors needs to stated their EP concentrations.

19. Authors should state their seeded plate (example 6 well plate or 60 mM dish or 24 well plate etc) and also cells density.

6. PLOS authors have the option to publish the peer review history of their article (what does this mean?). If published, this will include your full peer review and any attached files.

Reviewer #1: **Yes: **Wei Ling Lim

Reviewer #2: No

---

## [Author Response · Author response to Decision Letter 0]

6 Dec 2022

Dear Editors and Reviewers

Thank you very much for reviewing our manuscript and offering valuable advice.

We have addressed your comments with point-by-point responses and revised the manuscript accordingly.

Replies to Reviewers

Reviewer #1: 

The present manuscript by Luo and Sakamoto describes the neuroprotective effects of ethyl pyruvate (EP) against the 6-OHDA induced toxicity in SH-SY5Y cells, potentially mediated by the upregulation of autophagy. The current study appears similar to a recently published work by the same authors to some extent, including the in vitro model used, biochemical assays, presentation, etc. However, there are concerns in the manuscript that needs to be addressed and clarified.

Reply:

We appreciate for your constructive comments.

1. Materials and Methods – Perhaps provide a detail description on how 6-OHDA and EP is dissolved/prepared prior to the treatment. Was there any solvent/vehicle control included in the cell viability assay?

Reply:

Thank you for your comment. 6-OHDA was dissolved in dDW, and EP is a solution state. In this study, we used max 5mM EP, so we didn’t diluted it. We added this to the Materials and Methods section, please refer to Page 4 Line 17.

2. Method and Materials – For the cell apoptosis (Section 2.3) and ROS (Section 2.4) assays, the authors mentioned a post-treatment of 6-OHDA for only 6 hrs, which is different from the MTT assay and melanin content measurement. Why is there a difference in the 6-OHDA treatment among the different assays? Is this a typo or did the authors perform prior screening assays (different concentration and time-point) to justify the extent of toxicity induced by 6-OHDA in SH-SY5Y cells?

Reply:

Thank you for your comment. In our pervious study, we measured the different time-point, and we found that 6h treatment for ROS and apoptosis showed the best results. In this study we also used 6h treatment for ROS and apoptosis.

3. Results – Section 3.2 – The authors mentioned a significant increase in cell death induced by 6-OHDA treatment, which is only presented in the photomicrographs. Perhaps the authors should present the quantification data, since this is also mentioned in the Methods and Materials (Pg. 4) to quantify the apoptotic, necrotic and live cells. Furthermore, the fluorescence images presented in Figure 2a appears to be of poor quality and dim throughout, which should not be the case.

Reply:

Thank you for your comment. We have added the quantification data, please refer to Fig.2. The resolution of all Figures was increased to 550 dpi. Please refer to Fig.1-4.

4. Results – Section 3.3 – Similarly, measurement of the intracellular ROS levels should be presented instead of showing only the photomicrographs in the Results, as mentioned in the Methods section (Pg. 5).

Reply:

Thank you for your comment. We added the quantification data, please refer to Fig. 3.

5. There have been several prior studies on neuroprotective effects and mechanisms of EP in both in vitro and in vivo PD models. Perhaps the authors could improve the Introduction and/or Discussion of this manuscript to also discuss how this study fit into the current literature and strengthen the Discussion section.

Reply:

Thank you for your comment. We added more references into the Introduction and discussion section, please refer to Page 4 Line 3-9, Page 12 Line 10-13, Page 13 Line 11-16.

6. Minor grammatical errors present throughout the manuscript and please check thoroughly (E.g., Pg.2, L. 1.; Pg.6, L.18; etc).

Reply:

Thank you for your comment. We checked the manuscript again.　

Reviewer #2: 

This is an interesting manuscript examining the effect of ethyl pyruvate (EP) on 6-hydroxydopamine-induced SH-SY5Y cell apoptosis. Authors stated that EP could decrease cleaved caspase-3, pERK, and ERK protein levels, suggesting that EP reduces apoptosis via the ERK signaling pathway. Moreover, this study demonstrated that EP increased the protein levels of Beclin-1, LC-II and the ratio of LC-II/LC-I, suggesting that EP can upregulating autophagy. This is totlly a "descriptive" observation. However, still I have some concerns about the methods and data.The authors would have to provide additional experimental evidences about MAPK signaling pathway and autophagy as well as have multiple rising questions. Authors should provide the answers of those questions.

Reply:

Thank you for your constructive comments.

Comments:

1. Details method should need to be clarified by the authors. Example in Fig 1B, authors used 6-OHDA and EP but in method section autors did not explain anything about their drugs and also concentrations. The drug and concentrations should also be stated in the method part.Moreover, in MTT method authors added 10 % SDS solution, author needs to explain the reason. Author should provide the ref. of this method.

Reply:

Thank you for your comment. We added the relevant concentrations in the Method and Materials section, please refer to Page 4 Line 16-17. We used 10% SDS as a formazan solubilization solution and we added the relevant references, please refer to Page 5 Line 4.

2. In Fig. 3B, it seemed that pERK and ERK did not match with the quantitative values. There were some details that need to be clarified. The authors should provide the all raw protein bands of three independent experiments.

Reply:

Thank you for your comment. The quantitative values in Figure 3B are the ratio of pERK to ERK. we observed a decrease in pERK, but at the same time, the level of ERK also decreased significantly, so the ratio of pERK to ERK finally resulted in a large increase compared to the control group. raw data is attached to the additional images, please refer to Supplementary Figure.

3. mTOR promotes anabolic metabolism and inhibits autophagy induction. Therefore, the regulation of autophagy with mTOR inhibitors provides a new therapeutic strategy for a variety of diseases. Thus, in this study author should observe the expression of mTOR complaxes such as mTORC1 and mTORC2 and also evaluate the effect of EP on mTORC1 and mTORC2.

Reply:

Thank you for your comment. We gratefully appreciate for your valuable suggestion. Currently, we are unable to conduct experiments because our lab is being moved. Therefore, we added some ref about EP and mTOR. 

The previous studies showed the relationship between EP and mTOR, and EP decreases mTOR phosphorylation and thus promotes autophagy. We added this section to the discussion. Please refer to Page 13 Line 11-16.

4. Because authors focused on autophagy, thus to make a more influential study, authors should use Chloroquine and rapamycin in this study. Because Rapamycin promoted autophagy by blocking the mTOR pathway, and chloroquine enhanced apoptosis by blocking autophagy. Authors need to compare the effect of EP and Chloroquine and rapamycin on autophagy.

Reply:

Thank you for your comment. We gratefully appreciate for your valuable suggestion. Currently, we are unable to conduct experiments because our lab is being moved. Therefore, we added some ref about EP and Chloroquine and rapamycin.

The previous studies showed that rapamycin can restore increased mTOR activity (Gao et al., 2015). And chloroquine can block these protect effect of some medicine(He et al., 2022). We added this section to the discussion, please refer to Page 13 Line 11-15.

5. Authors should provide more information about 6-hydroxydopamine (6-OHDA) and Ethyl pyruvate (EP) in the introduction part. Example how 6-OHDA developes PD model? Are there any mechanism link with this disease?

Reply

Thank you for your comment. We added more references into the Introduction section, please refer to Page 3 Line 11-14,16-17, Page 4 Line 3-6.

6. Authors need to explain why they selected only ERK among the three proteins of MAPK pathway. How ERK plays an important role in this study? For more clarification authors should check p38 and JNK.

Reply:

Thank you for your comment. ERK and autophagy have a strong relative and have a strong connection with ROS, so we have mainly studied ERK in this experiment. About JNK and p38, previous study showed that JNK and p38 are not affected (Gómez-Santos et al., 2002), so in this experiment we didn’t measure JNK and p38 protein level. This was added in discussion section, please refer to Page 12 Line 10-13.

7. Authors stated that α-synuclein aggregation and oxidative stress in PD pathogenesis. Authors also stated the importance of α-synuclein in autophagy. Are there any reports about the α-synuclein and EP? If not, how authors explain this link between EP and α-synuclein?

Reply:

Thank you for your comment. The previous study showed that EP can decreases α-synuclein, and we added this to the discussion section, please refer to Page 13 Line 15-16

10. The authors should indicate the specific cat. number and dilution factor of the antibodies used in the Methods section to improve the reproducibility of their findings.

Reply:

Thank you for your comment. We added the cat. number and dilution factor of the antibodies in Method and Materials section, please refer to Page 6 Line 16-18, Page 7 Line 2.

11. Most of the images quality are very poor such as Fig 2A, 3A etc. Authors should provide high-quality images and blots.

Reply:

Thank you for your comment. The resolution of all Figures were increased to 550 dpi. Please refer to Fig.1-4.

12. Please check English grammar and sentence structure.

Reply:

Thank you for your comment. We checked the manuscript again.

13. Authors need to use molecular weight in the western blot images.

Reply:

Thank you for your comment. We revised it, please refer to Fig2-4.

14. In Fig 2 B cleaved caspase-3 showed 2 bands, Are there any reason, authors need to explain or provide another image?

Reply:

Thank you for your comment. The activation of Caspase-3 requires cleavage at Asp175, resulting in an activated p17/p19 protein fragment. And the purchased caspase-3 antibody is show 19kDa and 17kDa size bands at the cleaved caspase-3 position. Please refer to https://www.cellsignal.jp/products/primary-antibodies/caspase-3-antibody/9662

15. In western blot images Beta actin should be equal but in Fig 2B,3B and 4A why in all groups beta-actins are not equal?

Reply:

Thank you for your comment. We used BSA to determine the protein concentration of each sample, and when we did WB, used the same concentration. Some changes in other proteins may affect the final b-actin changes. To eliminate these different, we normalize the data by b-actin.

16. In western blot method, authors needs to explain about the dilution of antibodies. And also provide the how many percentage of gel authors used during WB?

Reply:

Thank you for your comment. We added the dilution factor of the antibodies and gel in Method and Materials section, please refer to Page 6 Line 13-18, Page 7 Line 2.

17. In discussion part authors demonstrated that Extracellular signal-regulated kinases (ERK1/2) are crucial regulators of neuronal responses associated with cell death [26] [27]. In several different 6-OHDA-induced cell death model, ERK1/2 activation also plays an important role [28][29]. Therefore, we measured the protein levels of pERK and ERK. Cells exposed to 6-OHDA showed a significant increase in ERK phosphorylation consistent with previous studies[24][26], and pretreatment with EP decreased the ERK phosphorylation. This suggests that EP reduces apoptosis via the ERK signaling pathway. But authors did not explain how 6-OHDA -induced cell death model ? and how to make link with ERK. Additionally, authors should provide some ref. about the EP and ERK in other diseases. In overall author should re-arrage the some paragraphs of discussion part and make it more clear.

Reply:

Thank you for your comment. We gratefully appreciate for your valuable suggestion. In the previous study, 6-OHDA-induced cell death involves ERK activation (Kulich and Chu, 2001), and in other model, EP treatment shown the effect on ERK phosphorylation(Lee and Kim, 2011) (Lee et al., 2012). We added more references into the discussion section, please refer to Page 12 Line 10-13. 

18. Apoptosis assay, ROS assay, Melanin Content first two lines are very similar, why? Authors needs to write in different ways. And in method section authors needs to stated their EP concentrations.

Reply:

Thank you for your comment. We revised it, and add the concentration of EP and 6-OHDA，please refer to Page 4 Line 16-17.

19. Authors should state their seeded plate (example 6 well plate or 60 mM dish or 24 well plate etc) and also cells density.

Reply:

Thank you for your comment. We revised it, please refer to Page 4 Line 15-18,

---

## [Editor Report · Decision Letter 1]

5 Feb 2023

Ethyl pyruvate protects SHSY5Y cell line against 6-hydroxydopamine-induced neurotoxicity by upregulation Autophagy.

PONE-D-22-21944R1

Dear Dr. Sakamoto,

We’re pleased to inform you that your manuscript has been judged scientifically suitable for publication and will be formally accepted for publication once it meets all outstanding technical requirements.

Kind regards,

Khuen Yen Ng, PhD

Academic Editor

PLOS ONE
---

## [Editor Report · Acceptance letter]

8 Feb 2023

PONE-D-22-21944R1 

Ethyl pyruvate protects SHSY5Y cells against 6-hydroxydopamine-induced neurotoxicity by upregulating autophagy 

Dear Dr. Sakamoto:

I'm pleased to inform you that your manuscript has been deemed suitable for publication in PLOS ONE. Congratulations! Your manuscript is now with our production department. 

Kind regards, 

on behalf of

Dr. Khuen Yen Ng 

Academic Editor

PLOS ONE